# Direct observation of ultrafast exciton localization in an organic semiconductor with soft X-ray transient absorption spectroscopy

D. Garratt [1✉], L. Misiekis[1], D. Wood[1], E. W. Larsen [1], M. Matthews[1], O. Alexander[1], P. Ye [1], S. Jarosch [1], C. Ferchaud[1], C. Strüber[1], A. S. Johnson [1], A. A. Bakulin [2], T. J. Penfold[3] & J. P. Marangos [1]

The localization dynamics of excitons in organic semiconductors influence the efficiency of charge transfer and separation in these materials. Here we apply time-resolved X-ray absorption spectroscopy to track photoinduced dynamics of a paradigmatic crystalline conjugated polymer: poly(3-hexylthiophene) (P3HT) commonly used in solar cell devices. The $\pi \rightarrow \pi^*$ transition, the first step of solar energy conversion, is pumped with a 15 fs optical pulse and the dynamics are probed by an attosecond soft X-ray pulse at the carbon K-edge. We observe X-ray spectroscopic signatures of the initially hot excitonic state, indicating that it is delocalized over multiple polymer chains. This undergoes a rapid evolution on a sub 50 fs timescale which can be directly associated with cooling and localization to form either a localized exciton or polaron pair.

[1] Quantum Optics and Laser Science Group, Blackett Laboratory, Imperial College London, London, UK. [2] Department of Chemistry and Centre for Processable Electronics, Imperial College London, London, UK. [3] Chemistry—School of Natural and Environmental Sciences, Newcastle University, Newcastle upon Tyne, UK. ✉email: d.garratt15@imperial.ac.uk

Energy capture through light induced processes in organic systems is at the heart of attempts to reduce our reliance upon unsustainable fuel sources using emerging clean energy technologies from artificial photosynthesis to organic solar cells. Photoexcitation of these molecular crystals and macromolecules generates tightly bound molecular excitons which are more localized and less prone to dissociation than excitonic states in inorganic semiconductors[1–4]. Despite this, efficient charge photogeneration can be achieved in organic bulk heterojunction solar cell devices[5,6]. Optical and theoretical studies on these materials have suggested that this is due to electronic and vibronic coupling phenomena promoting a delocalization of the initially excited state in the donor, which enhances the efficiency of charge transfer and separation[7,8]. This effect is pronounced in the π conjugated polymer poly(3-hexylthiophene) (P3HT), the subject of this work, where lamellae stacked crystalline regions in the thin film[9] enhance this delocalization between polymer chains[10,11]. After excitation, coupling of the electronic wavefunction to the vibrational and torsional modes of the polymer is assumed to lead to a rapid localization of the exciton[12–14] on ultrafast (sub 100 fs) timescales. This is detrimental for the efficiency of solar cell devices using this material, therefore developing experimental methodologies which can study the effect are crucial for optimizing the performance of these devices. However, probing these localization processes directly is not possible with existing ultrafast optical techniques[15,16] both due to the lack of sensitivity to the spatial structure of the excited state wavefunction and the extremely short timescales often involved.

X-ray absorption spectroscopy (XAS) is sensitive to the spatial extent of the valence wavefunction due to the localized nature of the core hole. Furthermore, the screening of the core-hole potential and therefore the core level energy is dependent on the oxidation state of the absorbing atom and leads to absorption features which strongly depend on the chemical environment of the atom. This sensitivity has been demonstrated with picosecond studies of photovoltaic materials using synchrotron radiation[17] but these X-ray sources lack the temporal resolution required to capture highly transient states often important in solar energy conversion. This has recently changed with the advent of femtosecond to attosecond soft X-ray sources from high harmonic generation[18–23], and free electron lasers[24]. Pioneering works using these sources have demonstrated time-resolved XAS spectroscopy (TR-XAS) of gas phase molecules at the carbon K-edge[25–29] and at the M and L edges of heavier elements in semiconductors and metals[30–33]. More recently, femtosecond x-ray photoemission spectroscopy has been applied to track charge separation in a organic heterojunction[34].

In this work we apply element specific, soft X-ray transient absorption spectroscopy to directly probe ultrafast exciton localization in a conjugated polymer. Extending TR-XAS to photoexcited organic semiconductor materials has so far proved technically challenging due to their high photosensitivity (leading to a low damage threshold), low thermal conductivity and the requirement to probe in the water window spectral range. We overcome this by using a high flux water window high harmonic source in combination with careful control of the pump pulse intensity and monitoring of sample damage and heating. With the aid of time-dependent density functional theory (TDDFT) calculations of the X-ray absorption spectrum to interpret our measurements, we find a TR-XAS spectral signature of exciton delocalization between polymer chains in P3HT and time resolve localization of the exciton. We explore the dependence of exciton delocalization on π stacking distance and confirm that the identified effect depends strongly on this parameter and therefore interchain coupling strength.

## Results

Figure 1a shows a schematic of the experimental methodology. Briefly, free standing P3HT films with uniform thickness are produced by spin coating. The thickness of the samples is optimized to ~100 nm to maximize the signal to noise ratio of the measurement. The visible absorbance spectrum of our P3HT samples is presented in Fig. 1b. The spectrum shows a broad multi-hump peak centered at 2.25 eV corresponding to the π→π* transition in P3HT. On the low energy side of the absorption band two peaks are observed consistent with previous measurements. These are related to interchain excitations and have a complex nature due to H- and J- aggregation as well as vibronic states as discussed in detail in references[11,35]. The samples are pumped with a 15 fs optical pulse, with a pulse energy of ~80 nJ, centered at the peak of the absorption band. An attosecond soft X-ray supercontinuum extending from ~150 to ~350 eV is then used to probe the ensuing dynamics at the carbon K-edge. Further details on the experimental setup can be found in the material and methods section.

A typical high harmonic spectrum is shown in Fig. 1c. Here the decrease in flux at around 284 eV is due to absorption due to carbon contamination of the X-ray optics and filters. The absorption spectrum of the P3HT sample is plotted below. We simultaneously resolved the sulfur $L_{2,3}$, $L_1$, and carbon K-edge at 165, 230, and 284 eV respectively. The spectra are in good qualitative agreement with those published in the literature[36,37]. The three peaks at the sulfur $L_{2,3}$ edge primarily correspond to spin–orbit split σ*(C–S) orbitals with very weak contributions from the optically excited π* state. Therefore this edge is not directly sensitive to electronic dynamics due to photoexcitation of the π→π* transition. Pump probe measurements, detailed in Supplementary Note 1, show that differential absorption signals at this edge are caused by pump induced heating of the samples and are therefore delay independent. We can exploit this and monitor the sample damage and heating at the sulfur L edge. Two pre-edge absorption peaks are observed at the carbon K-edge. The lowest energy peak at 285.5 eV corresponds to the C1s→π* transition, where the highest sensitivity to the optically excited dynamics is expected. The higher energy peak at 287.5 eV has contributions from σ*(C–S) orbitals as well as σ*(C–H) orbitals belonging to the alkyl side chains. This peak is therefore primarily sensitive to structural changes in the polymer. Above 290 eV the X-ray absorption is saturated due to a combination of background carbon contamination and the strong absorption of P3HT at these energies. This means that we do not resolve these peaks in the P3HT absorption spectrum and instead focus on the lower energy features.

We first present the results from pump probe studies at the carbon K-edge and then discuss how these relate to the ultrafast dynamics of the polymer. The differential absorption (ΔA) at the carbon K-edge following resonant photoexcitation of the π→π* transition is shown in Fig. 2a. Here, positive delays correspond to the X-ray pulse following the visible pump. From a singular value decomposition of the time-dependent trace[38], we identify three significant components in the data. At short time delays (<30 fs, shown in Fig. 2b) three differential absorption features are visible. A decrease in absorbance around the maximum of the C1s→π* absorption peak (285.5 ± 0.4 eV) is accompanied by a weak increase in absorption, 1.2 ± 0.5 eV below the peak (referred to as feature A) and a stronger increase in absorption 0.5 ± 0.4 eV above the peak (feature B). Feature A corresponds to transitions into the vacant π orbitals generated upon photoexcitation, while feature B corresponds to a narrowing and spectral blue shift of the C1s→π* peak giving rise to the derivative profile in the differential spectrum. The temporal dependence of the two positive differential absorption features is shown in Fig. 2b (bottom row).

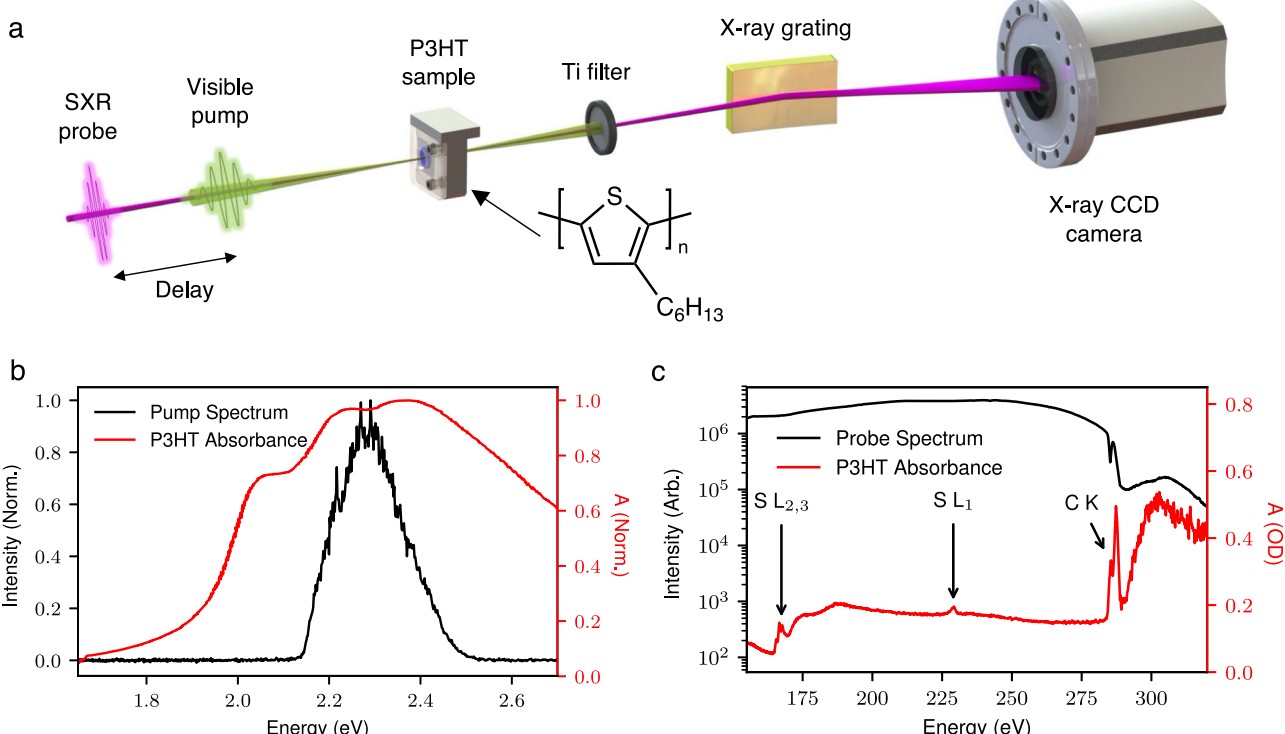

**Fig. 1 Experimental methodology and static absorption spectra of P3HT. a** The experimental setup for time-resolved soft X-ray spectroscopy. A 15 fs visible pump pulse excites the $\pi \rightarrow \pi^*$ transition in P3HT, and a temporally delayed attosecond soft X-ray pulse probes the carbon and sulfur absorption edges. **b** The visible absorption spectrum of the P3HT samples used in this work, and the pump pulse spectrum which is centered at the maximum of the $\pi \rightarrow \pi^*$ resonance in P3HT. **c** A typical soft X-ray spectrum extending to ~330 eV (black line). The red line is the X-ray absorption spectrum of the P3HT sample, absorption features at the sulfur $L_{1,2,3}$ and carbon K-edges are resolved simultaneously. The assignment of the absorption peaks are discussed in the main text.

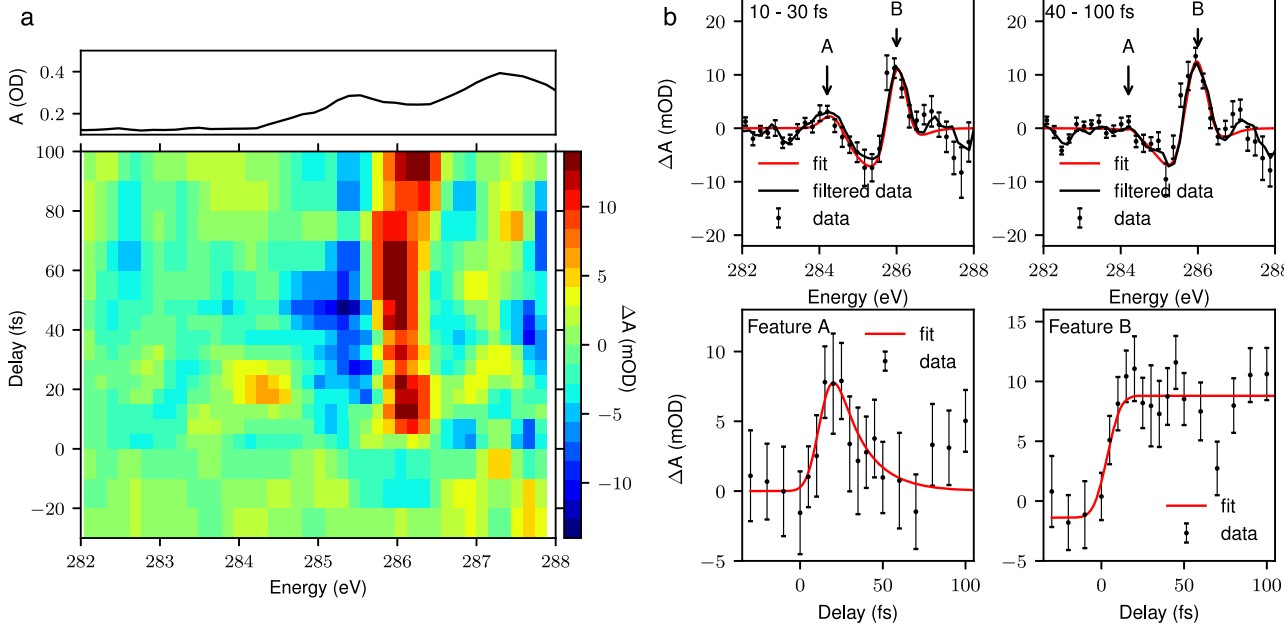

**Fig. 2 Experimental time-resolved X-ray absorption spectrum. a** TR-XAS spectrum of P3HT from −30 fs to 100 fs in the vicinity of the carbon K-edge. A static absorption spectrum of the sample is plotted above, on the same energetic axis. **b** The average TR-XAS spectrum across 10 to 30 fs and 40 to 100 fs (top row). The black lines show the measured spectrum and the red line shows the three component Gaussian fit to the data used to determine the spectral positions of the features discussed in the text. The bottom row shows the time evolution of the two positive differential absorption features A and B, discussed in the main text. The red lines show an exponential decay and step function fit to the temporal traces for feature A and B respectively. The error bars correspond to ±1 standard error. Further details can be found in the Methods section.

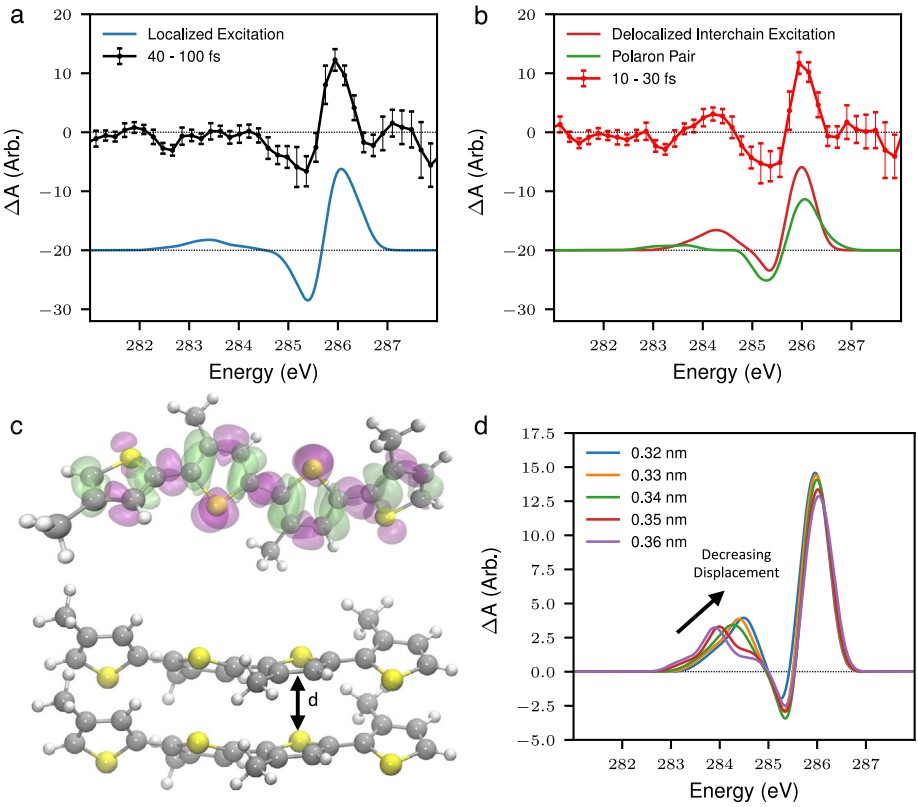

**Fig. 3 Calculated X-ray absorption spectra for excited state species in P3HT. a** Comparison of the differential absorption signal at between 40 and 100 fs with the theoretical prediction for a tetrathiophene oligomer which approximates the singlet exciton localized on a single polymer chain. **b** A comparison between the differential absorption signal at short time delays (10–30 fs) and the theoretical differential absorption spectrum for a tetrathiophene oligomer dimer, which approximates a delocalized interchain excitation the aggregated polymer and a constrained cationic–anionic tetrathiophene oligomer dimer, which models a polaron pair in the polymer. The error bars in the experimental traces shown in (**a**, **b**) correspond to ±1 standard error. Further details can be found in the methods section. **c** The electron density difference between the simulated ground and excited states in the tetrathiophene oligomer, purple indicates a loss of electron density and green indicates a gain indicating that there is a transfer of electrons from the green regions to the purple regions in the excited state relative to the ground state. Below is a ball and stick model of the simulated tetrathiophene oligomer dimer, showing the π stacking distance, d. **d** The simulated differential absorption spectrum for the tetrathiophene oligomer dimer as a function of π stacking distance. Transitions above 286 eV are above the ionization potential. As we have used atomic centered Gaussian basis sets, we expect these transitions around the continuum to be unconverged and therefore have not included a description of the nature of these features in the paper.

The temporal traces corresponding to features A and B are fitted with a Gaussian convolved with an exponential decay and a step function respectively. We find that feature A is highly transient and decays with a time constant of $16 \pm 8$ fs while feature B persists for the 100 fs of delay addressed in the measurement. The timescale of the A feature is close to instrument response limited and therefore the true decay time of the feature may be faster. The features are not fitted to a common time zero and we observe a small temporal offset of $9 \pm 4$ fs between the onset of feature B and feature A, this is again below the instrument response and close to the error in the fit. We therefore do not consider further effects below the pump pulse duration in the following discussion but these do merit further studies with shorter pump pulses.

## Discussion

To interpret the experimental data, we perform simulations of the photoexcited XAS spectrum of a model oligothiophene system consisting of a single tetrathiophene oligomer using Restricted Excitation Window Time-Dependent Density Functional Theory (REW-TDDFT)[39] within the approximation of the PBE0[40,41]

exchange and correlation functional as implemented within the ORCA quantum chemistry package[42] (further details provided in the theoretical methods section). Figure 3a (blue line) shows the simulated differential absorption spectrum for the lowest lying triplet excited state in this system, used to approximate the singlet exciton in P3HT, which has the same electronic structure as shown in Supplementary Table 1. This represents a good approximation due to the lack of spin sensitivity of XAS spectroscopy as shown in ref. [43]. Here the theoretical signal is calculated from the difference of the ground and excited state XAS spectra. Despite the small size of the simulated system with respect to polymer (consisting of hundreds of repeat units) remarkably good agreement is found between the theoretical prediction and the main feature observed in the experiment at time delays > 40 fs. This derivative profile, as outlined above, corresponds to an edge shift suggesting a reduction in charge density on the carbon atoms after photoexcitation. This observation is confirmed using the electron density difference, $\rho_{T1} - \rho_{S0}$ for the model which shows a transfer of electron density away from the carbon and towards the sulfur atoms (Fig. 3c). The spectral narrowing and blue shift of the C1s$\rightarrow\pi^{*}$ peak is therefore

well described by a model of the singlet exciton on a single polymer chain. The lack of significant time evolution of this peak across the 100 fs of delay addressed in the experiment is consistent with the ~200 ps[44] lifetime of the exciton from optical studies.

The theoretical transient spectrum shown in Fig. 3a contains a very weak positive differential absorption feature at 283 eV, which is at lower energy than feature A in the experimental data. The calculated spectra also show a weak dependence on the nuclear geometry of the oligomer in this region of the spectrum (see Supplementary Fig. 5) and therefore cannot be used to explain this feature.

There are three main processes which could be responsible for the discrepancy: Exciton–exciton annihilation, polaron pair generation and localization processes. At the excitation densities used in this work (0.29 nm$^{-3}$), the exciton–exciton annihilation rate is significant (~$0.7 \times 10^{13}$ s$^{-1}$), but not enough to give rise to the ~10 fs timescale of the feature. In addition, this process leads to the transition of one exciton to a higher lying excited state[45] and would therefore be expected to lead to increased X-ray absorption at lower energies to what is observed in the experiment. Polaron pair generation is also thought to occur on sub-100 fs timescales, and to be enhanced at high excitation densities. To investigate whether this would be consistent with the spectral features observed, we have modeled this by simulating the XAS spectrum of a tetrathiophene oligomer dimer in which one is constrained to be cationic and the other anionic, as described in the theoretical methods. As shown in Fig. 3b, it does not accurately reproduce the observed features at short time delays, either in intensity or relative energy position. This is especially true for the A feature, although a weak feature is present, it is not strong enough nor in the correct position to be consistent with the observed spectral shape. We note that the simulated polaron pair also contains a derivative profile in the differential absorption spectrum. We therefore cannot exclude the presence of species such as the polaron pair because it also has a spectral profile consistent with feature B in the experimental trace which is present at all positive time delays. This is consistent with experimental[8] and theoretical[14] studies on P3HT which reported polaron pair formation on sub 100 fs timescales.

Localization effects present in the aggregated polymer are not captured in the simulated system shown in Fig. 3a as it contains only a single oligothiophene chain and therefore the effect of interactions between lamellae stacked polymer chains on the electronic structure are not included. Although complete description of the X-ray absorption in the highly extended polymer is beyond current theoretical methods, we model aggregation effects with two tetrathiophene oligomers placed 0.34 nm apart (Fig. 3b). Crucially, although the shape of the derivative profile remains largely unchanged, this model leads to a significant shift in energy and increase in the intensity of the A feature, consistent with the experimental transient spectrum recorded at early times. This is because a delocalization of the excited electronic state between polymer chains (corresponding to interchain excitations) gives an enhancement in the transition amplitude for the X-ray probe below the edge. A similar effect is also observed for oligomer trimers and tetramers as shown in Supplementary Fig. 4. Our calculations indicate that this is due to a subtle change in the electronic structure of the dimer, leading to an increase electron density at the carbon 2p orbitals compared to the monomer, leading to extra strength for the 1s–2p dipole transition observed in the measurement. Consequently, while we cannot exclude the presence of species such as the polaron pair, it is only the interchain delocalization that gives rise to the A feature present on short timescales. This indicates that the A feature offers a direct measure of the extent of interchain delocalization

of the electron hole pair and its decay. We have found that this effect, and the differential absorption spectrum of the isolated oligomer does not depend strongly on the number of units as shown in Supplementary Fig. 3. This indicates that the intrachain localization does not produce the same strong changes in electronic structure and can therefore be distinguished in the TR-XAS signal.

The enhancement of the A feature in the simulated dimer shows a strong dependence on the separation in the π stacking direction. Our simulations show that smaller displacements increase the amplitude and shift the peak to higher energy. This is due to increased interchain interaction between the π stacked tetramers, as shown in the density difference plots in Supplementary Fig. 6, further enhancing the 1s–2p dipole transition (Fig. 3d). The separation which gives the best fit to the experimental data in the vicinity of feature A is ~0.34 nm, which is comparable the 0.38 nm reported π stacking distance in regioregular P3HT[46]. As is suggested by Fig. 3d, there will also be a distribution of stacking distances in the polymer, which will broaden the feature, with higher weighting in the amplitude coming from the smaller distances in the distribution. The strong dependence on π stacking distance suggests that the TR-XAS signal might also be sensitive to morphology of the polymer which is in turn important for optimizing charge generation efficiency.

Having established the connection between feature A and the extent of interchain delocalization, its temporal evolution, which decays with a time constant of 16 ± 8 fs, can be directly associated with the formation of an exciton localized on a single chain. This is likely driven by cooling of the initially formed of hot exciton[47] and is consistent with the timescale reported by recent quantum dynamics simulations for this material[14]. This is likely to be accompanied by the separation of the exciton into the polaron pair, which would be difficult to distinguish experimentally due to the similarity of its simulated transient spectrum with the localized exciton simulation. We now discuss our result in the context of the existing optical literature for P3HT. Ultrafast optical studies targeting localization processes[15,16] have observed a rapid Stokes shift in the time-resolved fluorescence signal accompanied by depolarization of the polymer fluorescence and absorption. These effects occurred below the timing resolution of the experiment (~100 fs) therefore the timescale could not be accurately determined. In addition, these observations indicate a rapid evolution of the excited state wavepacket but cannot be associated directly with localization and could also be due to structural changes. Here we observe a direct spectroscopic signature which can be associated with the extent of interchain delocalization in the polymer. This interchain delocalization observed here is also expected to lead to the formation of polarons/polaron pairs in neat P3HT. Optical studies have found that the efficiency of polaron pair generation on the order of 10% in regioregular P3HT[48]. At present we cannot distinguish polaron pair generation from the generation of a localized singlet exciton however subtle differences in their transient spectra may be able to be distinguished in future TR-XAS studies.

In conclusion, we have demonstrated the application of time-resolved XAS at the carbon K-edge to a prototypical crystalline homopolymer donor material in organic photovoltaics. We have observed the TR-XAS signature of the singlet exciton in this system and shown that the signal is sensitive to delocalization of the exciton between polymer chains. By combining this with the exceptional timing resolution of our soft X-ray and visible pump source, we have directly observed the localization of the interchain exciton in P3HT within 20 fs to form either a exciton localized on a single chain or a charge separated polaron pair with the electron and hole localized on distinct polymer chains. This

sub-20-fs timescale matches well the dynamics of vibronic coupling to high frequency vibrations such as the C=C bond stretch and is in agreement with theoretical studies showing that bond length alterations, characteristic of exciton self-trapping, may occur on a 20-fs timescale[49]. At the same time, this timescale is much faster than molecular torsions and coupling to the environment, indicating their minor contribution to exciton localization. These localization processes are driven by vibronic coupling of the electronic states in the polymer to the vibrational and torsional modes of the system. Our simulations suggest that the TR-XAS signal is also sensitive to nuclear motion in the polymer (see Supplementary Fig. 5) and therefore future studies with moderate improvements in the signal to noise ratio will be able to unravel the role of vibronic coupling in the localization dynamics on femtosecond timescales. The influence of polymer morphology on the localization dynamics observed here could be further explored in future work by varying the regioregularity of the polymer to reduce the crystallinity of the samples and therefore the interchain coupling. This would also reduce conjugation length on a given polymer chain, allowing for the influence of this to be investigated further. The technique presented here is generally applicable to organic photoconversion materials and could also be extended to study the effect of localization charge transfer dynamics in Donor:Acceptor bulk heterojunction solar cells such as PM6:Y6[50] or P3HT:PCBM, as well as photosynthetic light-harvesting complexes. This could be achieved by probing at different elemental absorption edges, for example the K-edges of nitrogen and oxygen, to resolve the localization of charge at different sites in the system. Our results therefore open up a range of new possibilities in studying the influence of localization on charge transfer in organic photoconversion systems.

## Methods

**Experimental methods**. The soft X-ray source and optical pump pulse used in this experiment are derived a commercial optical parametric amplifier, pumped with 8 mJ, 30 fs pulses from a Ti:Saphire laser system. The idler, which has a central wavelength of 1750 nm, is compressed to ~12 fs using a hollow core fiber pulse compression system giving ~600 µJ pulses with an excellent spatial profile. This is split into a pump and probe pulses using an annular mirror with a 1 mm hole. The majority of the pulse energy is reflected and used for soft X-ray high harmonic generation and ~30 µJ is transmitted and used to generate the pump pulse. In the pump arm, the delay between the pump and probe is controlled with a piezoelectric stage with 1 nm accuracy. The visible pump pulse is then generated via cascaded second harmonic generation and sum frequency generation in a single 100 µm thick beta barium borate (BBO) crystal. A half waveplate before the generation crystal ensures that the polarization of the pump pulse and soft X-ray probe are parallel. After the generation crystal, the third harmonic is spectrally filtered by 4 reflections from low dispersion dielectric mirrors before recombination with the soft X-ray probe at an annular mirror with a 3 mm hole. In the probe arm, the portion of the beam reflected by the annular mirror is focused with a 15 cm focal length $CaF_2$ lens into a 1 mm length gas cell filled with 2.5 bar of Ne, housed in a double differential pumping jacket, generating a soft X-ray supercontinuum extending to ~330 eV. The driving laser field is then blocked by a 200 nm Ti filter and the soft X-ray beam passes through the center of the annular recombination mirror. The pump and probe are focused onto the target using a 50 cm Au coated toroidal mirror to focal spot radii of $54 \pm 1$ µm and $40 \pm 2$ µm respectively. The X-ray spectrum is then spectrally dispersed by a concave, aberration corrected grating, with an effective line density of 1200 l/mm, onto a thermoelectrically cooled CCD camera. The focal point of the harmonic radiation constitutes the source point for this spectrometer. A second 200 nm Ti filter, at the entrance to the spectrometer blocks the residual pump pulse not absorbed in the sample. The energy range of the spectrometer extends from 150 eV to 350 eV with an estimated spectral resolution ($\Delta E/E$) of $1.4 \times 10^{-3}$. The detected soft X-ray flux at 280 eV is ~$3 \times 10^4$ counts/s/eV corresponding to an estimated flux at generation of $1.85 \times 10^6$ ph/s/1%bw, in line with previously reported results using similar systems[18]. We also confirm from the carrier envelope phase dependence of the harmonic cut-off that the soft X-ray pulse at this photon energy comprises an isolated pulse of likely sub-femtosecond duration[18]. While the duration of the soft X-ray probe is not measured directly, streaking measurements reported in the literature[51] indicate that high harmonic generation driven by few-cycle short wavelength infrared fields produces soft X-ray pulse durations on the order of 100 s of attoseconds.

Temporal overlap between the pump and probe pulses is determined via spectral interferometry between the pump pulse and the third harmonic of the HHG driving laser generated in the gas cell. This enables time zero to be determined to an accuracy of 15 fs.

**Sample preparation**. The free standing P3HT samples are produced by a bilayer deposition method. A clean glass substrate is first spin coated with a thin (150 nm) water soluble layer of polystyrenesulfonic acid (PSSA) which is dried for 10 min at 60 °C. Regioregular P3HT (molecular weight = 52 kDa), dissolved in dichlorobenzene with a concentration of 20 mg/ml is then spin coated on top of the water soluble layer at a spin speed of 1000 rpm. After drying for 10 min at 60 °C, the samples are submerged in water to dissolve the PSSA layer and the P3HT layer is transferred to a stainless steel sample holder with 25, 200 µm drilled holes for transmission measurements. The thickness of the samples used in the measurements is estimated to be ~100 nm from the X-ray absorption of the sample

**Data analysis**. The static P3HT absorption spectra presented in Fig. 1 are measured by integrating the soft X-ray spectrum for 5 min with and without the sample in the beam. The sample absorbance (A) is then calculated from the sample transmission spectrum $I_s(E)$ and the background spectrum, $I_0(E)$ according to the Beer Lambert law, such that

$$A(E) = -\log_{10}\left(\frac{I_s(E)}{I_0(E)}\right). \tag{1}$$

Transient absorption spectra are obtained by recording P3HT transmission spectra with and without the pump pulse for a series of pump probe time delays. Each transmission spectrum is integrated for 1 min. The pumped and unpumped X-ray transmission spectra are normalized to the harmonic flux at 250 eV and then the differential absorbance is calculated according to

$$\triangle A(E) = -\log_{10}\left(\frac{I_p(E)}{I_{up}(E)}\right), \tag{2}$$

where $I_p$ and $I_{up}$ denote the sample transmission with and without the pump pulse. The signal is averaged over 15 repeated delay scans, and the error bars shown in Figs. 2, 3 and Supplementary Fig. 1 correspond to ±1 standard error and are estimated from the standard deviation of the repeated measurements. A two-dimensional Wiener filter is applied to the transient absorption spectrum to obtain the plots shown in Fig. 2. The temporal traces are obtained by integrating 1 eV regions centered at each differential absorption peak. The integrated yield of the negative differential feature, is subtracted from each yield to give the temporal traces.

**Pump pulse characterization**. The pump pulse duration is characterized via second harmonic frequency resolved optical gating (SH-FROG)[52]. Supplementary Fig. 7 shows the measured and reconstructed FROG spectrograms along with the reconstructed temporal and spectral intensity and phase of the pump. The measurement gives a pump pulse duration (FWHM) of 16 fs. There is some residual third order spectral phase which gives the small pre/post pulse observed in the temporal intensity profile. This is attributed to the third order phase accumulated by the fundamental due to transmissive optics before the generation crystal.

The pump focal spot size is measured to be $54 \pm 1$ µm ($1/e^2$ radius) from the transmission of the pump beam through a pinhole placed in the sample plane. This leads to a pump fluence of $870 \pm 60$ µJ cm$^{-2}$ and an average excitation density across the probed sample volume of 0.29 nm$^{-3}$. At the excitation densities employed here, exciton–exciton annihilation is thought to be a significant deactivation mechanism for the photoexcited exciton. Assuming an exciton–exciton annihilation rate coefficient of $2.3 \times 10^{-8}$ cm$^3$ s$^{-1}$[53], we estimate that this process reduces the singlet exciton lifetime from 200 ps[44] to $0.6 \pm 4.4 - 0.35$ ps, but this remains well outside our measured delay range and is not captured in these measurements.

**Sample damage and heating**. P3HT is a highly photosensitive material with low thermal conductivity (~0.2 Wm$^{-1}$ K[54]). Therefore, particular attention was paid to avoiding permanent damage to the sample by the pump pulse train and to characterizing the sample temperature during the measurement. The damage threshold of the P3HT samples used in this work was determined to be $1.1 \pm 0.1$ mJ cm$^{-2}$. Above this pump fluence, photobleaching of the samples is observed. We also see a reduction in the magnitude of the absorption features at the sulfur L$_{2,3}$ edge and carbon K-edge indicating a structural change in the polymer. For the data presented in the main text (Fig. 2) we confirm that significant sample damage does not occur by comparing the sulfur L$_{2,3}$ X-ray absorption before and after the measurement (see Supplementary Note 2 for details).

Due to the low thermal conductivity, heat conduction from the sample is not efficient so elevated sample temperatures are reached when irradiated by the pump pulse. To estimate the sample temperature during the pumped acquisitions, we simulate the temperature distribution of the sample for CW illumination of equivalent power and focal spot size (corresponding to 52 µW of absorbed heat flux). The simulations include thermal conduction from the pumped volume and

to the sample holder as well as black body cooling from the faces of the sample. Supplementary Fig. 8 shows the results of the temperature simulations. The average sample temperature across the probed volume of the sample is 180 °C. This is below the melting temperature for P3HT, and we do not see any permanent damage to the samples after performing the pump probe measurements. The elevated temperature during the pumped acquisitions is reversible and manifests as a delay independent background in the transient absorption spectrum. This is estimated by taking the average of the transient spectrum at negative time delays and subtracting this from all other delay points.

**Theoretical methods**. All simulations were performed using the ORCA quantum chemistry package[42]. The XAS spectra were simulated using TDDFT adapted for the core-hole spectra[39]. All calculations were performed within the approximation of the PBE0 exchange and correlation (x-c) potential[40,41], using def2-TZVP[55] basis set. Throughout a fine integration grid was applied and the energy change convergence criteria associated with the self-consistent field cycle was set to $10^{-9}$ a.u. The TDDFT equations were solved for 20 states, within the Tamm Damcoff approximation and the interaction with the X-ray field was described using the electric quadrupole approximation. The P3HT model consisted of 4 repeat units, i.e., a tetrathiophene oligomer. The dimer was modeled by duplicating the initial structure and displacing it between 0.32 and 0.36 nm along the z-axis, corresponding to the π stacking direction. All calculated spectra were broadened using a Gaussian with a full-width half maximum of 0.5 eV. The transient spectrum was generated by subtracting the spectrum of the electronic ground state from a spectrum calculated using the lowest triplet state reference and shifting the energetic axis by 9.7 eV for the single oligomer and polaron pair and 10.1 eV for the oligomer dimer to best match the experimental trace. The lowest triplet state is used to approximate the singlet exciton in P3HT, which has the same electronic structure. This represents a good approximation due to the lack of spin sensitivity of XAS spectroscopy as shown in ref. [43]. The orbital character of the excited states in the tetrathiophene oligomer and tetrathiophene oligomer dimer are detailed in Supplementary Tables 1, 2. The polaron pair was simulated generating a dimer of the tetrathiophene oligomer separated by 34 nm. The reference ground state was constrained so that one of the oligomers is anionic and the other cationic. The REW-TDDFT spectrum was calculated for this reference state using DFT(PBE0) level of theory using a def2-TZVP basis set. The transient spectrum was generated by subtracting the ground state spectrum from this calculated spectrum.

## Data availability

The experimental and theoretical data generated in this study have been deposited in the Zenodo database (https://doi.org/10.5281/zenodo.6525897).

## Code availability

The codes used in the experimental data analysis and theoretical simulations are available from the corresponding author upon request.

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

## Acknowledgements

We gratefully acknowledge invaluable scientific discussions with Jenny Nelson and James Durrant. Assistance from Chris Brahms, Dane Austin, Alvaro Sanchez-Gonzalez, and Emma Simpson on the HHG source and the early development of the experimental apparatus are gratefully acknowledged as is the invaluable technical assistance from Andrew Gregory and Susan Parker. We acknowledge the UK Engineering and Physical Sciences Research Council for the funding of this work principally through grant numbers EP/R019509/1, EP/T006943/1, EP/V026690 and the MURI (EPSRC/DSTL) grant EP/N018680/1, and for earlier stages of the work EP/I032517/1 and the ERC Advanced Grant "ASTEX". D.G. and O.G.A. acknowledge support from the EPSRC grant EP/L016524/1. S.J. acknowledges support from the Marie Sklodowska-Curie grant agreement no. 641272. A.S.J. and D.A.W. also acknowledge support from Marie Curie Initial Training Network EC317232. A.S.J. acknowledges support from Natural Sciences and Engineering Research Council of Canada (NSERC) PGSD3-454096-2014. A.A.B. acknowledges support from the Royal Society and Leverhulme Trust. A portion of the results presented in this paper were previously reported in a conference proceedings[56].

## Author contributions

D.G., L.M., D.W., E.W.L., M.M., O.A., P.Y., C.F., S.J., A.S.J., and C.S. performed the experiments and developed the apparatus. D.G. analysed the data. J.P.M. conceived the experiment and supervised the implementation. T.J.P. performed the simulations. A first draft of the paper was written by D.G. with assistance from J.P.M., A.A.B., and T.J.P.. All authors contributed to the final draft of the paper.

## Competing interests

The authors declare no competing interests.
