## [Peer Review File · Nature Communications]

REVIEWER COMMENTS

Reviewer #1 (Remarks to the Author):

The manuscript by Garratt et al reports a study of the initial ultrafast dynamics in P3HT free standing films by soft-Xray transient absorption spectroscopy. Following resonant photoexcitation of the polymer with a visible pump pulse, the first 100 fs dynamics at the carbon K edge are monitored to track ultrafast redistribution of the photoexcited electronic species with atom selectivity. Based on calculations of a model oligothiophene and its dimer, the main experimental observations are explained in terms of ultrafast exciton localization on a single polymer chain on a timescale of ~20 fs after optical excitation.

A deep microscopic understanding of the initial out-of-equilibrium dynamics in organic photoactive materials is fundamental towards control of those dynamics on the nanoscale. This is highly relevant not only in solar energy conversion, but also for other applications that rely on light-initiated ultrafast charge transport processes. As such this study could be of large interest for a broad audience.

However, there are some issues to address before a final decision on publication can be made. Below are my main concerns and suggestions.

- The soft-Xray spectrum covers the Carbon K edge and the Sulfur L1, L2,3. The authors report to have found the L edge features to be “primarily sensitive to photoinduced sample damage and ...heating...which do not manifest as delay dependent changes”. In the methods section, they refer to control measurements monitoring potential damage at the S L2,3 edge before and after the measurement. I think these measurements should be reported, e.g., in the supporting information.

- To rationalize the experimental observations, the authors perform TDDFT simulations based on (i) an oligothiophene chain to isolate intrachain excitations, and (ii) a dimer to address interchain excitation of the lamellar-like aggregate structure forming in the ordered P3HT domains.

- The physical meaning of the electron density difference reported in Fig 3C should be briefly explained for a broader audience.

This density difference should show “a transfer of electron density away from the carbon and towards the sulfur atoms” to explain the derivative-like feature. It is challenging to verify this from

the figure. I suggest to report also the ground and excited state densities separately, in the supporting information.

- The authors have tested whether polaron pair generation would be consistent with the observed features by simulations of “cationic and anionic P3HT hexamers, as described in the supporting information”. I cannot find any description of those simulations in the supporting information.

Polaron pairs are charge-neutral excitations in which spatially separated electrons and holes interact via their Coulomb interaction and are each coupled to their lattice distortion. In the simulations in Fig 3B the spectrum of a “cationic hexamer” is used to analyze the signature of polaron pairs in the XAS. This is quite confusing since it suggests charged quasiparticles instead. Please clarify this point.

Please also explain how molecular vibrations, that are critical role for polarons, are accounted for in the present theoretical approach.

- The study of the dimer with varying distance shows a weaker peak between ~ 286 eV and ~ 287 eV whose intensity increases with decreasing distance. Can the authors comment on the possible origin of this peak?

- In the conclusions the authors write that the interchain exciton localization within 20 fs is “driven by vibronic coupling of the electronic states in the polymer to the vibrational and torsional modes of the system”. This sentence is not justified with enough experimental or theoretical data. It also seems to me somehow in contrast with the previous statement, a few lines above, that polaronic excitations “do not seem to show polaron/polaron pair formation within the first 100 fs”. Since this is potentially very important and relevant in view of the applications in devices, it requires more careful explanations, also in the context of the current understanding on the role of vibronic couplings for the excited state dynamics in P3HT in the literature.

Minor: Thiophene hexamer or oligothiophene would be more appropriate than “P3HT hexamer” when referring to the theoretical results.

Reviewer #2 (Remarks to the Author):

In this manuscript, carbon K-edge transient absorption, with high time resolution is used to study ultrafast dynamics in P3HT. More specifically sub 50 fs dynamics are observed which are attributed to localization of an initially hot delocalized excited state, which is backed up by theory. While the ability to perform TR-XAS in the carbon K-edge with high harmonic generation sources is relatively recent, this technique has been widely demonstrated before, and the quality (S/N) of the results is not very impressive when compared to less dense and more challenging targets (in gas phase or solution) usually studied. However, the time resolution (sub 20 fs) of the experiment is impressive and certainly challenging. In my opinion, the work was well conceived and the results are concretely backed up by the data. The discussion is interesting and certainly raises important points that might be of general interest to the community.

I have a few comments/concerns, listed below:

1. One of the most interesting aspects of the experiment was the ability to resolve both the carbon K- and sulfur L-edges on the same spectra. However, the authors claim that no photoinduced dynamics (other than damage) were observed in the sulfur site. Is this expected? Given the large extent of the excitation changes, I would imagine that the sulfur atom would also see meaningful electronic density changes that should show up as a transient signal. It would be more convincing if the authors showed, with theory and better reasoning, that no signal was expected at the sulfur edge in the first place.
2. The decay of feature A is fit to a time constant of 16 ± 8 fs. This seems to me to be time resolution limited, which means the true process might be even faster. If that is the case, please mention it in the text. What are the implications of an even faster decay? Is it possible to push the time resolution to probe the process beyond the IRF of the current experiment?
3. The authors claim a small temporal offset of 9 ± 4 fs between the onset of features B and A. Given the S/N of the measurements, the time resolution of the experiment and the uncertainty in time zero determination, is this offset really meaningful? Its implications are also not expanded on, so I am not sure what it means in the context of the discussion.
4. The sentence "This is likely driven cooling of the initially formed of hot exciton" is missing a "by", it should read "This is likely driven by cooling..."
5. Methods section: What is the estimated photon count at ~ 300 eV? Is a single attosecond pulse used for the probe? It was not clear from the description of the experiment. If that is not the case, what is the expected time resolution of the soft-xray probe?

Review of Nature Communications manuscript NCOMMS-21-40749

“Direct Observation of Ultrafast Exciton Localization in an Organic Semiconductor with Soft X-ray Transient Absorption Spectroscopy”

by Douglas Garratt et al.

This manuscript reports on the direct experimental observation, using soft X-ray transient absorption spectroscopy, of exciton localization in regioregular poly(3-hexylthiophene) (P3HT), on a sub-50 fs time scale. Experiments are performed with an optical pump pulse of 15 fs duration followed by an attosecond soft X-ray pulse at the carbon K edge, leading to the identification of two dominant features around the C $1s \rightarrow \pi^*$ transition (285.5 eV) during the first 30 fs: First, a blue shift and narrowing of the absorption edge (“feature B”) and second, a transient red-shifted signal (“feature A”) that is attributed to transitions into the vacant π orbitals. The extremely fast decay of feature A, within 16 fs, is taken as a signature of localization of an initially delocalized exciton.

The interpretation of the data relies on theoretical calculations of the differential absorption spectrum, shown in Figure 3 together with the experimental data. These calculations employ TDDFT with the PBE0 functional, adapted to the computation of core hole spectra (see Ref. [39]). The spectra are computed for the lowest lying triplet excited state which is used to approximate the singlet exciton in P3HT. Calculations are performed for a hexamer oligothiophene chain and for stacked dimers with varying inter-chain distance. From the single-chain calculations, it is inferred that a reduced charge density on the carbon atoms leads to the blue shift of the C $1s \rightarrow \pi^*$ transition (feature B). As for feature A, these calculations only show a weak feature in the relevant spectral range. In contrast, the dimer calculations exhibit a stronger spectral signatures corresponding to feature A, along with a dependence on the inter-chain distance, suggesting that this feature is indeed a signature of delocalization. The authors further argue that exciton-exciton annihilation and polaron pair formation are not responsible for this spectral signal.

These are highly interesting observations that would provide a first direct demonstration of ultrafast exciton localization in regioregular P3HT, in line with recent quantum dynamical studies (Ref. [14]). Hence, the manuscript is *a priori* suitable for publication in Nature Communications. However, I am not entirely convinced that the theoretical analysis provides unequivocal evidence for the authors’ interpretation. Unfortunately, the description of the computational results is very sparse. Hence, several issues should be clarified, as detailed in the following. A suitably revised manuscript should be acceptable for publication in Nature Communications.

The following issues should be addressed in a revised manuscript version:

1. The differential absorption spectrum is computed from the lowest lying triplet excited state – while the optically excited state is a bright singlet state. Hence, energies are shifted by as much as 69.95 eV. It is argued that “the lowest triplet state is used to approximate the singlet exciton in P3HT, which has the same electronic structure.” Ref. [43] is cited to justify this approximation; however, this reference addresses intersystem crossing in 2-thiopyridone – a very different system. From my viewpoint, the assumed equivalence of the lowest triplet state and the optically excited singlet needs to be demonstrated by electronic

structure calculations for a P3HT oligomer at an appropriate level of theory. Computational results should be reported in detail in the Supporting Information.

2. As for the calculations for the stacked dimer species, the authors do not specify which electronic state is considered, and whether the optically bright exciton is approximated by a triplet analog. Again, details on the computations should be provided in the Supporting Information. Even for a dimer, the correct description of delocalized excitonic states is non-trivial (see, e.g., Li et al., *J. Chem. Theory Comput.* 2014, 10, 3280).

3. As pointed out by the authors, XAS spectra were also computed for charge-separated species, i.e., cationic and anionic hexamer species. Again, details on the relevant electronic states and the level of computation should be provided in the Supporting Information.

4. Assuming that photoexcitation of regioregular P3HT can be treated within an H-aggregate model (see Ref. [35]), a spectral shift of the optical $\pi - \pi^*$ absorption should be observed that provides a signature of the initial exciton delocalization. Is such a shift observed in the initial optical excitation, possibly in line with results by conventional spectroscopy for regioregular vs. regiorandom P3HT? Is it possible to estimate the degree of initial delocalization?

5. Again referring to the H-aggregate description, photoexcitation at the upper excitonic band edge is known to be followed by ultrafast internal conversion, eventually leading to a – delocalized – dark exciton. This type of process, which does not necessarily lead to immediate localization, could be reflected in the observed X-ray absorption signals. From the theoretical analysis, it is not quite clear to me how this process could be distinguished from the authors' interpretation in terms of localization. In fact, in the quantum dynamical study of Ref. [14], internal conversion on a time scale around 20 fs is found to precede localization.

6. The page number of Ref. [43] should be corrected: *Chem. Eur. J.* 2019, 25, 1733–1739.

7. Apparently, some of the results reported in the manuscript were previously published as a conference proceedings paper which does not seem to be cited; hence, the relevant reference should be added:

D. Garratt et al., "Ultrafast Exciton Dynamics in Poly(3-hexylthiophene) Probed with Time Resolved X-ray Absorption Spectroscopy at the Carbon K-edge," 2021 Conference on Lasers and Electro-Optics Europe & European Quantum Electronics Conference (CLEO/Europe-EQEC), 2021, pp. 1-1, doi: 10.1109/CLEO/Europe-EQEC52157.2021.9542778.

We thank the reviewers for their careful reading of the manuscript and insightful comments, below we detail a point by point response. In the following, the responses are in blue and the relevant changes to the manuscript are included in italics

Reviewer #1

The manuscript by Garratt et al reports a study of the initial ultrafast dynamics in P3HT free standing films by soft-Xray transient absorption spectroscopy. Following resonant photoexcitation of the polymer with a visible pump pulse, the first 100 fs dynamics at the carbon K edge are monitored to track ultrafast redistribution of the photoexcited electronic species with atom selectivity. Based on calculations of a model oligothiophene and its dimer, the main experimental observations are explained in terms of ultrafast exciton localization on a single polymer chain on a timescale of ~20 fs after optical excitation.

A deep microscopic understanding of the initial out-of-equilibrium dynamics in organic photoactive materials is fundamental towards control of those dynamics on the nanoscale. This is highly relevant not only in solar energy conversion, but also for other applications that rely on light-initiated ultrafast charge transport processes. As such this study could be of large interest for a broad audience.

However, there are some issues to address before a final decision on publication can be made. Below are my main concerns and suggestions.

- The soft-Xray spectrum covers the Carbon K edge and the Sulfur L1, L2,3. The authors report to have found the L edge features to be “primarily sensitive to photoinduced sample damage and ...heating...which do not manifest as delay dependent changes”. In the methods section, they refer to control measurements monitoring potential damage at the S L2,3 edge before and after the measurement. I think these measurements should be reported, e.g., in the supporting information.

Response: We are grateful to the reviewer for raising this point and agree this information should be added. We have added an additional section to the supplementary materials detailing the control measurements performed at the sulfur L edge. We reiterate that there is not expected to be a spectroscopic signature of the exciton at the S L edge (in contrast to the C K edge), but monitoring this edge has played a key role in our ability to characterize and ameliorate thermal induced changes from the pump pulse. We also include spectra showing the effect of pump induced sample damage at this edge. The changes to the supplementary material are as follows:

“Measurements at the Sulfur L Edge

Here we present the results from an additional set of measurements performed at the sulfur L edge. Since the absorption cross section at this edge is significantly lower than that at the carbon K edge, these measurements were performed with thicker samples (approximately 200 nm) to optimize the signal to noise ratio at this edge. The Ti filters used in the C K edge measurements to filter the fundamental driving field after the harmonic generation and to block the pump beam before the spectrometer were replaced with Zr in order to increase the flux at around 165 eV. The low transmission of the Zr filters above ~210 eV precludes recording the carbon K edge absorption spectrum simultaneously in these measurements.

At the sulfur $L_{2,3}$ edge there are three pre-edge absorption features which primarily correspond to spin orbit split sulfur $2p \rightarrow \sigma^*(C-S)$ transitions with a very weak contribution (<3%) to the highest energy peak from a sulfur $2p \rightarrow \pi^*$ transition (1). This can be understood as a consequence of atomic dipole selection rules: In a molecular orbital picture, the π^* orbital is primarily a combination of sulfur $3p_z$ and carbon $2p_z$ orbitals. Since the sulfur $2p \rightarrow 3p$ transition is dipole forbidden any absorption amplitude at this edge is due to the weak contribution from sulfur $3d$ orbitals to the π^* orbital. Therefore, this edge is not strongly sensitive to electron density changes at the sulfur site following photoexcitation of the $\pi \rightarrow \pi^*$ transition at 2.25 eV. Instead, we mainly expect any differential absorption signal at this edge to reflect structural/geometrical changes in the polymer in the vicinity of the sulfur atoms.

The raw pumped and unpumped transmission spectra of P3HT at the sulfur L edge are shown in Supplementary Fig. 1A along with the corresponding differential change in absorption (Supplementary Fig. 1B). Here the shaded regions correspond to 1 standard error in the differential absorption. In the pumped spectra we observe a reduction in amplitude and broadening of the absorption peaks, giving rise to three positive and three negative differential absorption features. The time evolution of the differential absorption spectrum is between -200 and +300 fs shown in Supplementary Fig. 1C along the time dependence of the differential absorption signal integrated across each of the positive peaks (Supplementary Fig. 1D). Within the signal to noise ratio of the measurement, there are no significant time dependent changes in the differential absorption signal and the signal persists at negative time delays. This therefore strongly suggests that the signal is due to reversible heating of the samples by the pump pulse. Simulations of the pump induced temperature change presented in Supplementary Fig. 5 indicate that the steady state temperature of the samples is 180 °C during the pumped acquisitions. The heating induced broadening therefore dominates the signal at this edge because, as discussed above, it is insensitive to photoexcitation.

Supplementary Fig. 1. Sulfur $L_{2,3}$ edge measurements. (A) Pumped and unpumped X-ray transmission spectra at the sulfur $L_{2,3}$ edge at zero time delay. The shaded regions indicate 1 standard error in the measurement. (B) The corresponding differential absorption spectrum at zero time delay. (C) The differential absorption spectrum for time delays between -200 and +300 fs in 100 fs steps. The dashed lines indicate the integration regions used in panel D. (D) The integrated signal across each positive peak in the differential absorption spectrum as a function of time delay.

Sample Damage Control Measurements

In addition to being sensitive to thermal heating of the samples by the pump pulse, the sulfur L edge is also sensitive to long term damage of the samples. Supplementary Fig. 2A shows sulfur $L_{2,3}$ edge X-ray absorption spectra recorded before and after exposure to pump beam with a fluence of $1300 \pm 60 \mu\text{J}/\text{cm}^2$, above the sample damage threshold, for 60 minutes. The spectra are normalized to the magnitude of the $L_{2,3}$ absorption edge. The three absorption peaks broaden and become less distinct indicating a structural change in the polymer. The pump pulse fluence used for these tests was the maximum available in the current setup, but it is reasonable to assume that increased pump intensities will lead to a further broadening and decrease in absorption feature magnitude. This perhaps due to a reduction in geometrical order in the films, however further investigation into the microscopic damage mechanisms is required. Similar effects, although more dramatic, have been observed in the sulfur $L_{2,3}$ edge absorption spectra of bithiophene monolayers deposited on Ag surfaces after exposure to high intensity, white synchrotron radiation, and was attributed to a polymerization reaction which would form disordered layers (2). Supplementary Fig. 2B shows the sulfur $L_{2,3}$ edge absorption spectrum recorded before and after the measurements detailed in the main paper. While the noise level in these spectra is higher, a damage induced broadening of all peaks is not observed, indicating that the samples have not undergone significant damage.

Supplementary Fig. 2. Sulfur $L_{2,3}$ edge sample damage controls (A) The X-ray absorption spectra of P3HT samples at the sulfur $L_{2,3}$ edge before and after exposure to a pump pulse fluence of $1300 \pm 60 \mu\text{J}/\text{cm}^2$ for 60 minutes. (B) Sulfur $L_{2,3}$ edge absorption spectra recorded before and after the measurements presented in the main paper (Fig. 2).”

- To rationalize the experimental observations, the authors perform TDDFT simulations based on (i) an oligothiophene chain to isolate intrachain excitations, and (ii) a dimer to address interchain excitation of the lamellar-like aggregate structure forming in the ordered P3HT domains.

- The physical meaning of the electron density difference reported in Fig 3C should be briefly explained for a broader audience.

This density difference should show “a transfer of electron density away from the carbon and towards the sulfur atoms” to explain the derivative-like feature. It is challenging to verify this from the figure. I suggest to report also the ground and excited state densities separately, in the supporting information.

Response: We agree with the reviewer that discerning the transfer of electron density away from the carbon and towards the sulfur atoms from Figure 3c was difficult due to the opaque surfaces. However, we find that plotting the overall electron density of the ground and excited states is not the best solution as the overall change is small, only one electron changes orbital and this would be impossible to discern by inspection. Consequently, we have replotted the density difference in Figure 3c using transparent surfaces which more clearly show the loss of electron density primarily focused on the carbon atoms, which moved onto the sulfur atoms or into the bond region between carbons. We have also added the following to the caption of Fig 3 to better explain the physical meaning of the electron density difference:

“The electron density difference between the simulated ground and excited states in the thiophene hexamer, purple indicates a loss of electron density and green indicates a gain indicating that there is a transfer of electrons from the green regions to the purple regions in the excited state relative to the ground state.”

- The authors have tested whether polaron pair generation would be consistent with the observed features by simulations of “cationic and anionic P3HT hexamers, as described in the supporting information”. I cannot find any description of those simulations in the supporting information.

Response: The referee is correct and to clarify the details of the simulations we have added the following to the ‘Theoretical Methods’ section in the paper:

“The polaron pair was simulated by summing the spectral calculations for an isolated cationic and anionic thiophene hexamer. Both the anionic and cationic forms were optimised at DFT(PBE0) level of theory using a def2-TZVP basis set. Throughout we neglect the coulomb interaction between the cationic and anionic species meaning this corresponds to a first order approximation of the polaron pair. This combined spectrum, which approximates a non-interacting electron and hole on separate chains, was used as the excited state spectrum and the transient spectrum generated by subtracting the ground state spectrum.”

Polaron pairs are charge-neutral excitations in which spatially separated electrons and holes interact via their Coulomb interaction and are each coupled to their lattice distortion. In the simulations in Fig 3B the spectrum of a “cationic hexamer” is used to analyze the signature of polaron pairs in the XAS. This is quite confusing since it suggests charged quasiparticles instead. Please clarify this point. Please also explain how molecular vibrations, that are critical role for polarons, are accounted for in the present theoretical approach.

Response: As clarified in the additional description in the Theoretical Methods section described above, we have accounted for the lattice distortion by optimising the geometry of the cationic and anionic thiophene hexamer used in the simulated excited state spectrum.

However, we have neglected the Coulomb coupling between the cation and anion, and this therefore corresponds to a first order approximation of the polaron pair.

- The study of the dimer with varying distance shows a weaker peak between ~286 eV and ~287 eV whose intensity increases with decreasing distance. Can the authors comment on the possible origin of this peak?

Response: The origin of this transition is very diffuse orbitals associated with the electron being transferred into states around or slightly above the continuum. As we have used atomic centred Gaussian basis sets, we expect these transitions around the continuum to be poorly described and have not included a description of the nature of this feature in the manuscript.

To clarify this we have added the following to the caption of Figure 3d:

“Transitions above 286 eV are above the ionisation potential. As we have used atomic centred Gaussian basis sets, we expect these transitions around the continuum to be unconverged and therefore have not included a description of the nature of these features in the manuscript.”

- In the conclusions the authors write that the interchain exciton localization within 20 fs is “driven by vibronic coupling of the electronic states in the polymer to the vibrational and torsional modes of the system”. This sentence is not justified with enough experimental or theoretical data. It also seems to me somehow in contrast with the previous statement, a few lines above, that polaronic excitations “do not seem to show polaron/polaron pair formation within the first 100 fs”. Since this is potentially very important and relevant in view of the applications in devices, it requires more careful explanations, also in the context of the current understanding on the role of vibronic couplings for the excited state dynamics in P3HT in the literature.

Response: We agree with reviewers' concern and we have clarified the point as follows:

“This sub-20-fs timescale matches well the dynamics of vibronic coupling to high frequency vibrations such as the C=C bond stretch and is in agreement with theoretical studies showing that bond length alterations, characteristic of exciton self-trapping, may occur on a 20-fs timescale (49). At the same time, this timescale is much faster than molecular torsions and coupling to the environment, indicating their minor contribution to exciton localization.”

Minor: Thiophene hexamer or oligothiophene would be more appropriate than “P3HT hexamer” when referring to the theoretical results.

Response: We agree and have changed this throughout the manuscript.

Reviewer #2

In this manuscript, carbon K-edge transient absorption, with high time resolution is used to study ultrafast dynamics in P3HT. More specifically sub 50 fs dynamics are observed which are attributed to localization of an initially hot delocalized excited state, which is backed up by theory. While the ability to perform TR-XAS in the carbon K-edge with high harmonic generation sources is relatively recent, this technique has been widely demonstrated before,

and the quality (S/N) of the results is not very impressive when compared to less dense and more challenging targets (in gas phase or solution) usually studied. However, the time resolution (sub 20 fs) of the experiment is impressive and certainly challenging. In my opinion, the work was well conceived and the results are concretely backed up by the data. The discussion is interesting and certainly raises important points that might be of general interest to the community.

I have a few comments/concerns, listed below:

1. One of the most interesting aspects of the experiment was the ability to resolve both the carbon K- and sulfur L-edges on the same spectra. However, the authors claim that no photoinduced dynamics (other than damage) were observed in the sulfur site. Is this expected? Given the large extent of the excitation changes, I would imagine that the sulfur atom would also see meaningful electronic density changes that should show up as a transient signal. It would be more convincing if the authors showed, with theory and better reasoning, that no signal was expected at the sulfur edge in the first place.

Response: While we do see significant electron density changes at the sulfur site as shown in figure 3 C, the sulfur L_{2,3} edge is not sensitive to these due to selection rules governing the X-ray probe step. The change in electron density is due to photoexcitation of the $\pi - \pi^*$ transition and therefore for the probe to be sensitive to these changes it must have significant transition amplitude to these orbitals. At the sulfur L_{2,3} edge edge, there are three spin orbit split peaks corresponding to sulfur 2p - σ^* (C-S) transitions (J. Chem. Phys. 147, 244301 (2017)), where the final σ^* (C-S) orbital is at significantly higher energy than the photoexcited π^* orbital. In a molecular orbital picture, the π^* orbitals of thiophene are a combination of atomic sulfur 3p_z and carbon 2p_z orbitals. Since the sulfur 2p - 3p transition is dipole forbidden, the S 2p - π^* transition is also expected to be very weak. We note that there is some disagreement in the literature as to the strength of the S 2p - π^* contribution in the monomer. The most recent study (J. Chem. Phys. 147, 244301 (2017)), supported by TD-DFT simulations suggests that the sulfur 3d contribution to the thiophene π^* orbital is around 3%.

While the sulfur 2p - σ^* (C-S) peaks are not strongly sensitive to electron density changes, they are dependent on the geometrical/structural of the polymer in the vicinity of the sulfur site. These are expected to occur after photoexcitation due to vibronic coupling, but our experimental results at this edge do not show time dependent changes (see the updated supplemental material) and the response is independent of time delay and persists at negative delays. Therefore we assign these changes to reversible heating of the sample by the pump pulse.

The argument based on selection rules presented above does not hold for the L₁ edge which corresponds to an atomic 2s - 3p transition. However, this transition sits on top of a strong background of scattering from the L_{2,3} edge and therefore the absorption feature is weak. Any changes at this peak were below the experimental noise level. This could potentially be overcome by using thicker polymer samples, but the SNR would be limited due to the large difference in the absorption cross section of the X-ray probe and the visible pump. We note that probing at the sulfur K edge at around 2.5 keV would be a very interesting avenue for future studies as this would be sensitive to the changes in electron density at the sulfur site due to photoexcitation of the $\pi - \pi^*$ transition (although this is outside of the photon energy range available to our HHG source).

We have added an additional section to the supplementary material detailing these measurements (see above) and have made the following change to the main text:

“Therefore this edge is not directly sensitive to electronic dynamics due to photoexcitation of the $\pi \rightarrow \pi^$ transition. Pump probe measurements, detailed in the Supplementary Notes, show that differential absorption signals at this edge are caused by pump induced heating of the samples and are therefore delay independent.”*

On the subject of S/N compared to gas phase measurements we would suggest that the limits are not set solely by sample density, but in the case of solid state samples pump induced thermal effects and accumulated damage become of critical importance. Thus our measured S/N is at the state-of-the-art for a time-resolved X-ray absorption measurement in these samples. As explained above the precautions taken to ensure damage did not affect our measurements are now explained further in the supplemental material.

2. The decay of feature A is fit to a time constant of 16 \pm 8 fs. This seems to me to be time resolution limited, which means the true process might be even faster. If that is the case, please mention it in the text. What are the implications of an even faster decay? Is it possible to push the time resolution to probe the process beyond the IRF of the current experiment?

Response: We agree that the time constant is likely to be instrument response limited. Currently the instrument response is limited by the duration of the pump pulse which is measured to be \sim 15 fs with a transform limited pulse duration of \sim 10 fs. In the current setup the pulse duration could therefore be reduced via improved dispersion management, or by implementing novel generation methods for the generation of few femtosecond visible pulses for example (Nature Photonics volume 13, pages 547–554 (2019)). We have included the possibility of a faster timescale in the text:

“The timescale of the A feature is close to instrument response limited and therefore the true decay time of the feature may be faster.”

3. The authors claim a small temporal offset of 9 \pm 4 fs between the onset of features B and A. Given the S/N of the measurements, the time resolution of the experiment and the uncertainty in time zero determination, is this offset really meaningful? Its implications are also not expanded on, so I am not sure what it means in the context of the discussion.

Response: The offset between the onset of the two features is below the timing resolution of the experiment. Thus we do not expand on this point, even though we agree it is an apparent feature in the data. We have made this clear in the revised text:

“The features are not fitted to a common time zero and we observe a small temporal offset of 9 \pm 4 fs between the onset of feature B and feature A, this is again below the instrument response and close to the error in the fit. We therefore do not consider further effects below the pump pulse duration in the following discussion but these do merit further studies with shorter pump pulses.”

4. The sentence "This is likely driven cooling of the initially formed of hot exciton" is missing a "by", it should read "This is likely driven by cooling..."

Response: This has been corrected in the revised manuscript.

5. Methods section: What is the estimated photon count at ~ 300 eV? Is a single attosecond pulse used for the probe? It was not clear from the description of the experiment. If that is not the case, what is the expected time resolution of the soft-xray probe?

Response: We estimate that the photon flux of the HHG source at 300 eV is $\sim 2 \times 10^5$ ph/s/1%bw at generation corresponding to a detected count rate of $\sim 10^3$ photons/s. This reduction from the generated flux is due to significant losses due to carbon contamination and the low efficiency of the X-ray optics. The flux at generation is the same order of magnitude as previous studies from our and other groups using few cycle pulses at 1800 nm (Nature Communications 7, 1-6 (2016), Science Advances 4, eaar3761 (2018)).

While the pulse duration of the soft X-ray probe is not measured directly, full three dimensional simulations of soft x-ray HHG driven by short wavelength infrared fields indicate that close to the cut-off, the X-rays are emitted as an isolated attosecond pulse with duration on the order of 100s of attoseconds for the vast majority of CEP values. The measured CEP dependences of the cut-off in Science Advances 4, eaar3761 (2018) are thus strong evidence for an isolated sub-fs pulse. The ~ 100 as pulse duration have been experimentally confirmed via streaking measurements reported in (PRX 7, 041030 (2017)) for a similar set-up. In our experiment we are using high harmonic generation in neon and the x-ray source has a cut-off of around 330 eV, close to the carbon K edge.

We have now modified the experimental methods section to include an estimate of the photon flux at 280 eV (close to where the spectral signals are observed in the experiment) and the expected pulse duration of the soft X-ray probe:

“The detected soft X-ray flux at 280 eV is approximately 3×10^4 counts/s/eV corresponding to an estimated flux at generation of 1.85×10^6 ph/s/1%bw, in line with previously reported results using similar systems (18). We also confirm from the carrier envelope phase dependence of the harmonic cut-off that the soft X-ray pulse at this photon energy comprises an isolated pulse of likely sub-femtosecond duration (18). While the duration of the soft X-ray probe is not measured directly, streaking measurements reported in the literature (51) indicate that high harmonic generation driven by few cycle short wavelength infrared fields produces soft X-ray pulse durations on the order of 100s of attoseconds.”

Reviewer #3

This manuscript reports on the direct experimental observation, using soft X-ray transient absorption spectroscopy, of exciton localization in regioregular poly(3-hexylthiophene) (P3HT), on a sub-50 fs time scale. Experiments are performed with an optical pump pulse of 15 fs duration followed by an attosecond soft X-ray pulse at the carbon K edge, leading to the identification of two dominant features around the C $1s \rightarrow \pi^*$ transition (285.5 eV) during the first 30 fs: First, a blue shift and narrowing of the absorption edge ("feature B") and second, a transient red-shifted signal ("feature A") that is attributed to transitions into the vacant π orbitals. The extremely fast decay of feature A, within 16 fs, is taken as a signature of localization of an initially delocalized exciton.

The interpretation of the data relies on theoretical calculations of the differential absorption

spectrum, shown in Figure 3 together with the experimental data. These calculations employ TDDFT with the PBE0 functional, adapted to the computation of core hole spectra (see Ref. [39]). The spectra are computed for the lowest lying triplet excited state which is used to approximate the singlet exciton in P3HT. Calculations are performed for a hexamer oligothiophene chain and for stacked dimers with varying inter-chain distance. From the single-chain calculations, it is inferred that a reduced charge density on the carbon atoms leads to the blue shift of the C 1s $\rightarrow \pi^*$ transition (feature B). As for feature A, these calculations only show a weak feature in the relevant spectral range. In contrast, the dimer calculations exhibit a stronger spectral signatures corresponding to feature A, along with a dependence on the inter-chain distance, suggesting that this feature is indeed a signature of delocalization. The authors further argue that exciton-exciton annihilation and polaron pair formation are not responsible for this spectral signal.

These are highly interesting observations that would provide a first direct demonstration of ultrafast exciton localization in regioregular P3HT, in line with recent quantum dynamical studies (Ref. [14]). Hence, the manuscript is a priori suitable for publication in Nature Communications. However, I am not entirely convinced that the theoretical analysis provides unequivocal evidence for the authors' interpretation. Unfortunately, the description of the computational results is very sparse. Hence, several issues should be clarified, as detailed in the following. A suitably revised manuscript should be acceptable for publication in Nature Communications.

The following issues should be addressed in a revised manuscript version:

1. The differential absorption spectrum is computed from the lowest lying triplet excited state - while the optically excited state is a bright singlet state. Hence, energies are shifted by as much as 69.95 eV. It is argued that "the lowest triplet state is used to approximate the singlet exciton in P3HT, which has the same electronic structure." Ref. [43] is cited to justify this approximation; however, this reference addresses intersystem crossing in 2-thiopyridone - a very different system. From my viewpoint, the assumed equivalence of the lowest triplet state and the optically excited singlet needs to be demonstrated by electronic structure calculations for a P3HT oligomer at an appropriate level of theory. Computational results should be reported in detail in the Supporting Information.

Response: For the isolated monomer, the lowest singlet excited state is composed of a predominantly (97%) HOMO- \rightarrow LUMO transition. The lowest triplet state is 84% HOMO-LUMO. Therefore while they are not completely identical, they exhibit very similar electronic structure. We have included a table in the supporting information describing this.

2. As for the calculations for the stacked dimer species, the authors do not specify which electronic state is considered, and whether the optically bright exciton is approximated by a triplet analog. Again, details on the computations should be provided in the Supporting Information. Even for a dimer, the correct description of delocalized excitonic states is non-trivial (see, e.g., Li et al., J. Chem. Theory Comput. 2014, 10, 3280).

Response: For the stacked dimer system there are 4 close low-lying singlet states, which exhibit combinations of HOMO-1, HOMO, LUMO, LUMO+1, LUMO+2 transitions. The optically bright state is the S3 state, but the close proximity of the states means that rapid internal conversion down to the S1 state would be expected, as outlined in point 5 below. The lowest triplet state, used in the present work to approximate the excited state corresponds to a HOMO-1- \rightarrow LUMO transition (89%). The S1 and S3 states both exhibit mixed HOMO-1- \rightarrow

LUMO (50%) and HOMO->LUMO+1 (50%) character. These are very similar in both cases, suggesting that transitions from the bright exciton to the lowest singlet state would give rise to limited spectral changes. We have included a table in the Supplementary Discussion describing this.

3. As pointed out by the authors, XAS spectra were also computed for charge-separated species, i.e., cationic and anionic hexamer species. Again, details on the relevant electronic states and the level of computation should be provided in the Supporting Information.

Response: In response to referee one, we have provided a more detailed description of the charge-separated species. In both cases, we use the ground cationic and anionic state.

4. Assuming that photoexcitation of regioregular P3HT can be treated within an H-aggregate model (see Ref. [35]), a spectral shift of the optical $\pi \rightarrow \pi^*$ absorption should be observed that provides a signature of the initial exciton delocalization. Is such a shift observed in the initial optical excitation, possibly in line with results by conventional spectroscopy for regioregular vs. regiorandom P3HT? Is it possible to estimate the degree of initial delocalization?

Response: The HJ aggregate model of the optical absorption and fluorescence spectra of P3HT can be related to the initial exciton delocalization. In this model, spectral shifts in the absorption spectrum are due to vibronic selection rules governing the optical absorption and emission process. In a pure H-aggregate absorption is to the top of the exciton band and (0-0) vibrational peak in the emission spectrum is forbidden while in a J-aggregate absorption is to the bottom of the exciton band and the (0-0) emission transition is strongly allowed. Spano et al relate this model by considering isolated polymer chains to behave as J aggregated chromophores while interactions between pi stacked chains have H-aggregate character. Again this argument is based on dipole selection rules governing optical absorption and emission.

In our experiment, we are probing with carbon 1s to valence transitions and therefore the selection rules governing the light absorption process are different from the valence to valence transitions involved in optical absorption/emission. Therefore the same HJ aggregate model cannot be directly applied to X-ray absorption measurements.

We agree that it would in principle be possible to estimate the degree of delocalization from our measurement. Our simulations indicate that the strength of feature A is dependent on simulated pi stacking distance. Therefore, if this variable was fixed and the simulation was extended to include additional thiophene hexamer units, the extent of delocalization could in principle be extracted from the best fit to the experimental data. However, calculating the X-ray absorption spectra for such an extended system is beyond current theoretical capabilities.

5. Again referring to the H-aggregate description, photoexcitation at the upper excitonic band edge is known to be followed by ultrafast internal conversion, eventually leading to a delocalized - dark exciton. This type of process, which does not necessarily lead to immediate localization, could be reflected in the observed X-ray absorption signals. From the theoretical analysis, it is not quite clear to me how this process could be distinguished from the authors' interpretation in terms of localization. In fact, in the quantum dynamical study of Ref. [14], internal conversion on a time scale around 20 fs is found to precede localization.

Response: We agree that internal conversion may play a role in the observed dynamics. However it is unclear whether this process will strongly couple to the X-ray absorption spectrum where as the simulations presented in the paper indicate a clear spectral feature which can be associated with the localization of the exciton. There are also open questions as to the timescale of internal conversion and previous experimental works (*J. Phys. Chem. C* 2011, 115, 19, 9726–9739) suggest that slower timescales are associated with internal conversion.

6. The page number of Ref. [43] should be corrected: *Chem. Eur. J.* 2019, 25, 1733-1739.

Response: We thank the reviewer for highlighting this and it has been corrected in the revised manuscript.

7. Apparently, some of the results reported in the manuscript were previously published as a conference proceedings paper which does not seem to be cited; hence, the relevant reference should be added:

D. Garratt et al., "Ultrafast Exciton Dynamics in Poly(3-hexylthiophene) Probed with Time Resolved X-ray Absorption Spectroscopy at the Carbon K-edge," 2021 Conference on Lasers and Electro-Optics Europe & European Quantum Electronics Conference (CLEO/Europe-EQEC), 2021, pp. 1-1, doi: 10.1109/CLEO/Europe-EQEC52157.2021.9542778.

Response: We now reference this preliminary (1 page) abstract describing our work, we stress that the current manuscript goes much further in describing and reporting the experimental findings and the interpretation than that brief preliminary account.

Reviewer #1 (Remarks to the Author):

The authors have satisfactorily addressed my concerns. I have no further questions or comments. I recommend publication of the revised manuscript in Nature Communications.

Reviewer #2 (Remarks to the Author):

After the implementation of the modifications, I have no further comments and I can recommend the manuscript for publication in Nat. Comm.

Review of Nature Communications manuscript NCOMMS-21-40749A (revision), by Douglas Garratt et al.

In their revised manuscript version, the authors provide additional information in response to the referees, including the computational aspects of their study. Focussing on the latter aspect, I am unfortunately not convinced that the computational analysis is sufficiently detailed to support the conclusions drawn in the manuscript:

1. The Supplementary Tables 1 and 2 which have been added to the Supporting Information, provide information about the orbital character of the singlet and triplet transitions of the hexamer monomer and dimer species. Incidentally, the table headers indicate that the oscillator strength is provided, too – however, this is not the case such that the information on the bright state assignment is missing. Also, no discussion is given in the context of these tables. In their response letter, the authors state that “*The lowest triplet state, used in the present work to approximate the excited state corresponds to a HOMO-1-LUMO transition (89%). The S1 and S3 states both exhibit mixed HOMO-1-LUMO (50%) and HOMO-LUMO+1 (50%) character. These are very similar in both cases, suggesting that transitions from the bright exciton to the lowest singlet state would give rise to limited spectral changes.*” This is a very approximate argument, revealing that the electronic state character of the triplet state which was taken as reference significantly differs from the electronic character of the bright dimer state (whereas in the monomer case, the triplet and bright singlet dimer states are both dominated by the HOMO-LUMO transition). Also, information on the orbitals is missing, specifically regarding their localized or delocalized character. This is of key importance for the distance dependence shown in Fig. 3D of the manuscript. Given that accurate state-of-the-art calculations of similar excitonic dimer species have been reported in the literature (see, e.g., Li et al., J. Chem. Theory Comput. 2014, 10, 3280), the level of analysis by the authors unfortunately does not appear adequate. It is understood that the computation of X-ray spectroscopic signals is far from trivial, but the nature of the electronic states that are reached by the initial optical pulse is obviously crucial to understand the spectral feature (“feature A”) that the authors aim to analyse.

2. Regarding the spatial extension of the exciton, the authors respond “*We agree that it would in principle be possible to estimate the degree of delocalization from our measurement. Our simulations indicate that the strength of feature A is dependent on simulated pi stacking distance. Therefore, if this variable was fixed and the simulation was extended to include additional thiophene hexamer units, the extent of delocalization could in principle be extracted from the best fit to the experimental data. However, calculating the X-ray absorption spectra for such an extended system is beyond current theoretical capabilities.*” While I understand that the analysis is difficult, it

is clear from the authors' explanations that they do not attempt a quantitative analysis of exciton delocalization. This to some extent contradicts the statement in the abstract that delocalization across multiple polymer chains occurs. (*"We observe, for the first time, direct X-ray spectroscopic signatures of the initially hot excitonic state, indicating that it is delocalized over multiple polymer chains."*)

3. As for the calculations for polaron pairs, the authors added a paragraph explaining that spectra were obtained by summing "*spectral calculations for an isolated cationic and anionic thiophene hexamer*". Given that these calculations entirely neglect the Coulomb interaction between the charged species – i.e., the dominant interaction which characterizes the polaron pair as a bound quasi-particle – this analysis remains very approximate. No computational details relating to these results are provided in the Supporting Information.

While I continue to think that the authors' observations are highly interesting, the theoretical analysis of the key spectroscopic feature in question – relating to exciton delocalization – does not seem convincing to me at the present stage. From this viewpoint, I unfortunately cannot give unequivocal support to the publication of the manuscript in Nature Communications.

Before turning to our detailed response we would like to thank the referee for their careful examination of our work and this stimulating challenge to our conclusions. Although we must stress that the main novelty and importance of our work is in the experimental methodology advances that we have made, i.e. in introducing time resolved K edge spectroscopy with a sub-20 fs time resolution to the study of exciton formation in an organic semiconductor, we have also striven to perform the highest quality theoretical analysis to aid in the interpretation of our measurements. As can be seen in the revised manuscript we have addressed all of the concerns and have provided additional improved calculations to further verify our original findings. Whilst our newer calculations reconfirm the origin of the transient feature A in terms of interchain exciton delocalisation on a very fast timescale we do find that the polaron pair formation may play a role in the observed feature B over longer timescales. We are thus grateful to the referee in prompting us to pursue the simulation further.

1. The Supplementary Tables 1 and 2 which have been added to the Supporting Information, provide information about the orbital character of the singlet and triplet transitions of the hexamer monomer and dimer species. Incidentally, the table headers indicate that the oscillator strength is provided, too – however, this is not the case such that the information on the bright state assignment is missing. Need to clarify this point on oscillator strengths in the corrected table. Also, no discussion is given in the context of these tables. In their response letter, the authors state that *“The lowest triplet state, used in the present work to approximate the excited state corresponds to a HOMO-1-LUMO transition (89%). The S1 and S3 states both exhibit mixed HOMO-1-LUMO (50%) and HOMO-LUMO+1 (50%) character. These are very similar in both cases, suggesting that transitions from the bright exciton to the lowest singlet state would give rise to limited spectral changes.”* This is a very approximate argument, revealing that the electronic state character of the triplet state which was taken as reference significantly differs from the electronic character of the bright dimer state (whereas in the monomer case, the triplet and bright singlet dimer states are both dominated by the HOMO-LUMO transition). Also, information on the orbitals is missing, specifically regarding their localized or delocalized character. This is of key importance for the distance dependence shown in Fig. 3D of the manuscript. Given that accurate state-of-the-art calculations of similar excitonic dimer species have been reported in the literature (see, e.g., Li et al., J. Chem. Theory Comput. 2014, 10, 3280), the level of analysis by the authors unfortunately does not appear adequate. It is understood that the computation of X-ray spectroscopic signals is far from trivial, but the nature of the electronic states that are reached by the initial optical pulse is obviously crucial to understand the spectral feature (“feature A”) that the authors aim to analyse.

The referee raises important points. Firstly, in addressing these concerns we note that both the density difference shown in Figure 3c and the associated character of the states shown in Tables S1 and S2 were for the hexamer, i.e. 6 connected thiophenes in a single oligomer chain. In addressing a later comment from the referee we have compressed this to a tetramer, i.e. 4 connected thiophenes in a single oligomer chain. This made it computationally accessible to compute the core-hole spectra of larger stacks of oligomers (3- and 4-) as discussed in point 3 below. This means that Figure 3c (new plot below) and the data in Tables S1 and S2 has to be altered to reflect the tetrameric structure. In doing this we also tightened the SCF convergence and adopted a finer integration grid for the wavefunction as this was found to influence the strength of the A feature, which is the focus of the present paper. All calculations and the methods section in the paper have been changed to reflect this.

Density difference map for the tetrathiophene oligomer used in the calculations throughout the revised manuscript.

Important in the context of the concerns raised by the referee, the corrected Tables S1 and S2 highlight that the lowest excited triplet and singlet state of both the monomer and dimer are dominated by HOMO-LUMO transitions. It is also noted that the S_1 energy of the tetrameric dimer also overlaps strongly with the energy of the pump pulse, shown in Figure 1b. This indicates that while still an approximation, the use of the lowest triplet state as the initial wavefunction upon which to perform the core-hole excitation for this tetrameric system is a reasonable approximation.

To justify the effect of contracting the hexamer to a tetramer, we have also included in the SI a new figure showing the core-hole difference spectrum as a function of the number of repeat units in the single oligomer chain:

Supplementary Figure 3 showing the differential absorption spectrum for a single oligomer chain as a function of monomer units.

The new data for Table S1 is:

	Energy, Oscillator Strength	Dominant Transition Character
T1	2.32	HOMO-LUMO (0.88)
S1	3.43, $f=1.389$	HOMO-LUMO (0.99)

And for table S2 (2 oligomer chains containing 4 repeat units separated by 34 nm).

	Energy (eV), Oscillator strength	Dominant Transition Character
T1	1.88	HOMO-LUMO (0.88)

T2	2.25	HOMO-LUMO+1 (0.6)
T3	2.48	HOMO-LUMO+2 (0.5)
T4	2.73	HOMO-LUMO+3 (0.5)
S1	2.37, f=0.001	HOMO-LUMO (0.99)
S2	2.99, f=0.080	HOMO-LUMO+1 (0.6)
S3	3.13, f=0.0142	HOMO-1-LUMO+1 (0.7)
S4	3.31, f=0.010	HOMO-LUMO+2 (0.99)

Regarding the orbital information, *i.e.* the extent of delocalization, and the importance for the distance dependence, we have added an additional new figure to the SI shown below. This shows the density difference of the lowest triplet state for the model dimer as a function of the separation, d , between the two oligomers. This clearly demonstrates that as the separation increases, the overlap between the density on each chain decreases, as expected given the exciton coupling will decay as d^{-3} .

Supplementary figure 6 highlighting the effect of varying the separation between tetrathiophene oligomer dimers on the overlap of the electron density difference.

2. Regarding the spatial extension of the exciton, the authors respond “We agree that it would in principle be possible to estimate the degree of delocalization from our measurement. Our simulations indicate that the strength of feature A is dependent on simulated π stacking distance. Therefore, if this variable was fixed and the simulation was extended to include additional thiophene hexamer units, the extent of delocalization could in principle be extracted from the best fit to the experimental data. However, calculating the X-ray absorption spectra for such an extended system is beyond current theoretical capabilities.” While I understand that the analysis is difficult, it is clear from the authors’ explanations that they do not attempt a quantitative analysis of exciton delocalization. This to some extent contradicts the statement in the abstract that delocalization across multiple polymer chains occurs. (“We observe, for the first time, direct X-ray spectroscopic signatures of the initially hot excitonic state, indicating that it is delocalized over multiple polymer chains.”)

Our initial objective was to demonstrate that the exciton was delocalized beyond a single oligomer. However, we agree with the reviewers assessment. Consequently, as described above we have compressed our initial hexamer oligomer to a tetramer, *i.e.* 4 connected thiophenes in a single oligomer chain, making it possible to also study the delocalization beyond a dimer to include the effect of a model trimer and tetramer. The results are shown below and included in the resubmitted manuscript show that further delocalization of the exciton (as indicated in the density difference plots) do not have a significant effect on the observations of the A feature made from the dimer model.

As an aside it is noted that the positive component of the derivative feature is missing in these large models (trimer and tetramers) due to the computational expensive of calculating sufficient number of excited states (>150) required to capture it for these systems. The focus of the analysis for these models should therefore be on the A feature.

Supplementary figure 4 detailing new simulations exploring the effect of increased the number of π stacked tetrathiophene oligomers on the exciton delocalization and its effect on the differential x-ray absorption spectrum in the vicinity of feature A.

3. As for the calculations for polaron pairs, the authors added a paragraph explaining that spectra were obtained by summing “spectral calculations for an isolated cationic and anionic thiophene hexamer”. Given that these calculations entirely neglect the Coulomb interaction between the charged species – *i.e.*, the dominant interaction which characterizes the polaron pair as a bound quasi-particle – this analysis remains very approximate. No computational details relating to these results are provided in the Supporting Information.

We agree with the reviewer that our calculation in the previous submission was a rather severe approximation for the polaron pair. In addition, the reviewer is correct that no computational details of the polaron pair calculations were provided in the Supporting Information as they were outlined in the main manuscript.

To address these concerns, we have performed new simulations in which the core-hole spectrum of the dimer with oligomers, separated by 0.34 nm is calculated when a positive and negative charge has been constrained on each fragment during the initial SCF optimisation wavefunction. These new simulations include the Coulomb interaction between the charged species and have replaced the previous ones in the resubmitted manuscript. These show a similar trend, and although a pre-edge transition now appears, it is weak and at lower energy than the transitions observed experimentally and therefore cannot account for these observations.

Finally, we stress, as noted in the resubmitted manuscript that we cannot exclude the presence of species such as localized excitons or the polaron pair because the calculated spectral signal exhibit a derivative profile consistent with the spectral shape observed experientially. However, in all models it is only the interchain delocalization that gives rise to the A feature as highlighted in the direct comparison with the experimental data below. We have updated the discussion and conclusions in our revised manuscript to reflect these new findings. However, the core insight of the paper: that time resolved X-ray spectroscopy is directly sensitive to delocalization in organic semiconducting polymers is still valid. We cannot distinguish in the current study localization to bound excitons or charge separated polaron pairs due to the similarities in their predicted XAS spectra. Future higher sensitivity XAS measurements are likely to resolve this question as there are small differences between these alternatives, but this will require further developments in x-ray flux and temporal resolution.

A comparison of the revised calculations for the isolated tetrathiphene oligomer, the tetrathiphene oligomer dimer and the updated polaron pair model with the experimental data between 10 and 30 fs.

While I continue to think that the authors' observations are highly interesting, the theoretical analysis of the key spectroscopic feature in question – relating to exciton delocalization – does not seem convincing to me at the present stage. From this viewpoint, I unfortunately cannot give unequivocal support to the publication of the manuscript in Nature Communications.

We thank the reviewer for their appreciation of the work within the manuscript and hope that our new simulations and corresponding changes address the concerns raised. We also would like to

stress that our measurements with a time resolved x-ray spectroscopy methodology with < 20 fs temporal resolution demonstrate important and unique insights into the local electronic dynamics associated with exciton formation on unprecedented timescales. This methodology is likely to have high impact in resolving questions of this nature across a wide class of materials and so the work has an importance beyond the case studied.

REVIEWERS' COMMENTS

Reviewer #3 (Remarks to the Author):

In their revised manuscript version, the authors' computational analysis has been significantly improved. Additional calculations were performed for larger oligothiophene stacks, while reducing the chain length of the oligomer units. These calculations permit a satisfactory analysis of the effects of delocalization on "feature A", essentially confirming the authors' previous observations for a dimer model. Further, these calculations lead to a modified interpretation of "feature B" which could be due to both polaron pair generation and the generation of localized singlet excitons. Besides, the reported singlet vs. triplet state characters reported in Tables S1 and S2 appear consistent for the new calculations, and the information in these tables is now complete. Finally, in the computations for polaronic species, the Coulomb interaction between the charged species has been included. Overall, the authors responded to all concerns raised in my previous report.

While certain aspects of the analysis remain open, I believe that the level of the computational treatment and theoretical interpretation is now adequate. The modifications made in the main text and Supplementary Material are appropriate and considerably strengthen the interpretation side. In view of the highly interesting experimental results reported in this paper, which will doubtlessly have a strong impact on the field, I recommend publication of the present manuscript version.